# A Tale of Two Cities: Unpacking the Success and Failure of School Street Interventions in Two Canadian Cities

**DOI:** 10.3390/ijerph191811555

**Published:** 2022-09-14

**Authors:** Laura E. Smith, Veronique Gosselin, Patricia Collins, Katherine L. Frohlich

**Affiliations:** 1Department of Geography and Planning, Queen’s University, Kingston, ON K7L 3N6, Canada; 2École de Santé Publique (ESPUM), Centre de Recherche en Santé Pulique (CReSP), Université de Montréal, Montreal, QC H3N 1X9, Canada

**Keywords:** child-friendly cities, school streets, social innovation, program implementation, realist evaluation, street-based interventions

## Abstract

One innovative strategy to support child-friendly cities is street-based interventions that provide safe, vehicle-free spaces for children to play and move about freely. School streets are one such innovation involving closing streets around elementary schools to vehicular traffic to improve children’s safety as they come and go from school while providing opportunities for children to play and socialize on the street. Launching these initiatives in communities dominated by automobiles is enormously challenging and little is known about why these interventions are successfully launched in some places but not others. As part of a larger research project called Levelling the Playing Fields, two School Street initiatives were planned for the 2021–2022 school year; one initiative was successfully launched in Kingston, ON, while the second initiative failed to launch in Montreal, QC. Using a critical realist evaluation methodology, this paper documents the contextual elements and key mechanisms that enabled and constrained the launch of these School Streets in these cities, through document analysis and key informant interviews. Our results suggest that municipal and school support for the initiative are both imperative to establishing legitimacy and collaborative governance, both of which were necessary for a successful launch.

## 1. Introduction

### 1.1. Overview 

The prevailing structure of North American cities privileges the needs and convenience of motorists above all other road users. This dominance of motorized vehicles is antithetical to child-friendly cities, constraining children’s opportunities to play on and occupy streets, and to move about freely within and between neighbourhoods. These constraints are compounded by intensive parenting practices, which limit allowances for children to play and move about unsupervised. Indeed, compared to previous generations, children today have limited capacity for independent mobility, spend less time engaged in outdoor free play and tend to do so much closer to home, and are less likely to use active transportation to move around their neighbourhoods [1,2,3,4,5,6,7]. 

Particularly noteworthy are the low rates of active school travel (AST) in North American cities [8,9,10]. The determinants of AST are multifactorial, including time pressures, distance to school, weather and topography, parental perceptions of stranger-danger, as well as injury risks to children (real and perceived) posed by encounters with motorists on the journey to and from school [11]. One structural issue that exacerbates the low rates of AST is the hazardous traffic conditions that typify the spaces outside primary school sites at the beginning and end of the school days [11,12]. These chaotic traffic conditions are created and reinforced by the high numbers of children being driven to school by parents on a regular basis, many of whom live within the active travel range of the school (i.e., less than 2 miles) [13]. 

School Streets represent one approach to promote more AST by eliminating motor vehicle congestion around primary school sites at the beginning and the end of the day to enable children to come and go from school safely and independently [14,15]. More specifically, School Streets involve closing the street adjacent to a school to any traffic at school arrival and dismissal times. This creates a car-free zone outside of schools before and after the school day. Beyond safety, School Streets offer numerous benefits to communities, including reduced noise and air pollution, and greater space for play and socialization [16,17,18]. There is growing enthusiasm for introducing these child-friendly programs in cities across Canada, yet little is known about why these programs can operate in some places and not others. Using a critical realist approach, the objective of this study was to document the contextual elements and key mechanisms that enabled and constrained the launch of two School Street programs, drawing on the experiences of two affluent neighborhoods in two Canadian cities. The findings from this study offer important insights for communities seeking to disrupt the dominance of the automobile to create child-friendly cities that support healthy child development.

### 1.2. Street Rebalancing for Child-Friendly Cities

According to UNICEF (2022), a child-friendly city “…is a city, town, community or any system of local governance committed to improving the lives of children within their jurisdiction…in which the voices, needs, priorities and rights of children are an integral part of public policies, programs and decisions” [19]. The prevailing design of automobile-dominated communities throughout North America contravenes many of UNICEF’s guiding principles of a child-friendly city, including acting in the *best interests of the child*, respecting children’s *inherent right to life, survival and development*, having *respect for the views of the child*, and minimizing barriers to children’s *equity and inclusion*. Thus, a child-friendly city is one that offers supportive physical environments for children to fully participate in society—to play, learn, and move freely and independently about their communities. Indeed, studies have shown that children who engage in outdoor free play and independent active transportation in their communities have higher levels of physical activity, self-esteem, self-confidence, emotional resilience, and greater capacities to manage risk and deal with uncertainty [20,21,22]. 

To create environments that promote children’s wellbeing, a growing number of cities are introducing innovative street-based interventions that minimize the risks posed by motor vehicles, while encouraging children to reclaim the street for free play and independent mobility [23]. The School Street intervention is one strategy originating in Europe that has become commonplace in the UK [17]. For instance, a School Street program that began as a pilot at five schools in the borough of Hackney, London has now spread to over 500 schools across the city and a comprehensive report suggests that these programs are achieving their objectives of reducing vehicular congestion around schools and enabling active school travel [16,24]. A growing number of communities across Canada have piloted or are in the process of piloting School Streets. These pilots have operated for varying lengths of time, from 1 day in Victoria, British Columbia [25], 4 days in Toronto, Ontario [18], to a full school year in Winnipeg, Manitoba [26], however, most Canadian communities have tended to test them for 1–2 months only in the spring (e.g., Vancouver, Mississauga, Markham, Hamilton) [27]. Despite the growth of School Street interventions in various countries, little is known about why these programs successfully launch and operate in some places and not others.

### 1.3. Evaluating School Street Implementation Using Critical Realism

With this paper, we attempt to “unpack” the necessary contextual elements that make the launch of such street-rebalancing interventions possible. Aligned with a critical realist epistemology, we understand the world to be real, existing and operating independently of our awareness or knowledge of it [28,29]. According to this perspective, the world is comprised of three things: objects, structures and generative mechanisms, all of them existing in the realm of the real. When diverse objects are assembled in varying configurations, or networks, through existing structures, they trigger certain generative mechanisms leading to actual events, in our case, the launch of the School Street. Since many real elements and actual events are not visible, it is impossible for us to empirically measure, observe, or even be aware of all of them. This is especially the case for many of the objects within the social sphere [29]. Our understanding as to what might have led to an event (an outcome) is therefore an inference about its cause—what real elements were present, how they were structured together and what mechanism this configuration triggered [30]. 

Our positionality regarding the importance of critical realist evaluation stems, in part, from our object of study; change in human activities in urban centers. Additionally, our School Street intervention was planned to engage stakeholders and local citizens in a participatory urban planning approach. This engagement was critical to the success of our interventions as participatory processes are particularly important when developing a concerted vision of the problem as well as the possible solution. Throughout this participatory process, children, parents, neighbors, municipally elected officials and staff, as well as school principals contributed to the interventions’ development. Indeed, the exact locations of School Streets, their frequency and duration, their rules of operation, the activities that would be offered (or not), the expected level of supervision from adults, etc., were meant to result from collective discussions ensuring that the interventions would be relevant to local needs, concerns and preferences. 

Participatory processes are unpredictable, however, and require particular types of evaluation frameworks that allow for such unpredictability, unlike a randomized control trial, for instance. Some researchers might consider such interventions to be too out of control to be rigorously evaluated as they cannot be entirely standardized across sites. Others might ponder as to how the results of such an evaluation could ever be useful. By adopting a critical realist approach, our evaluation permitted us to document the contextual elements and the key mechanisms that enabled and constrained the launch of these two School Streets. Realist evaluation provided us with theory-driven explanations of how complex programs like School Streets work within the context of their implementation [31,32], thus of infinite use to those wishing to also implement School Streets in other jurisdictions.

To advance our understanding of School Street implementation, therefore, our team designed a community-based intervention research project to evaluate the feasibility, mechanisms of action, and impacts of School Streets in two Canadian cities, Montreal, Quebec and Kingston, Ontario. In partnership with a non-profit community implementation partner in each city, two School Streets were planned for a period of one year during the 2021–2022 school year in two affluent neighborhoods. Despite both teams using comparable techniques of community engagement and mobilization, only the Kingston School Street intervention was successfully launched, offering a unique opportunity to study what worked and what did not work to achieve the desired outcome.

## 2. Materials and Methods

### 2.1. School Street Intervention Settings

The implementation phase of the School Streets started in 2019 in both cities and included site selection, stakeholder and community engagement and mobilization, as well as logistics planning. Implementation was led by partners responsible for this task in both cities, using the same participatory urban planning approach [33]. Participatory urban planning is the “effective participation of residents and users in the programming and design of a project” [33]. The central tenet of this approach is that collaboration between urban planning professionals and different users of the city make it possible to best meet the needs of communities. This approach was incorporated into the planning process as many studies evaluating the successful implementation of urban planning interventions emphasize the importance of using grassroots, community-based approach for successful program implementation [34,35,36]. More specifically, Canadian studies that focused on active school travel interventions identified that involving the school community in the intervention increases the likelihood of implementation success as it increases engagement and awareness of the interventions within the community [34,36]. 

For School Streets, the actors involved included school boards, school principals and staff, school parents, children, local residents, elected officials, municipal staff and occasionally police services. Table 1 shows a more detailed description of the Montreal Urban Ecology Centre’s (MUEC) participatory urban planning approach. The launch of both School Streets was intended for September 2021.

In Montreal, the launch of the School Street was intended to take place in a high-income community not far from downtown [37]. The neighbourhood is a small (3.9 km^2^) borough representing 1.4% of Montreal’s total population, but with the highest concentration of children between 0–14 years old in the city [37]. In the school neighbourhood, 39% of the residents use a private vehicle as their main mode of commuting to work, lower than the general population of Montreal where 50% depend on a vehicle for commuting [38]. This borough was selected for its high density of children living within a short distance from the primary school. The borough identified a one-way street on which two schools are located (one public and one private) as an ideal candidate for the School Street intervention due to the high levels of traffic congestion observed between the schools. The School Street was planned to run twice every school day, during morning arrival and dismissal time, every school day for the 2021–2022 school year. The operation of the School Street would be managed by community volunteers. 

In Kingston, the launch of the School Street was planned to take place in a small (1.33 km^2^), high-income neighborhood of Kingston [39]. Like in Montreal, this community has just 2% of Kingston’s population [39], but one of the highest concentrations of children 0–14 at 16% of the census tract population [39]. In this neighborhood, around 48% of residents use a private vehicle as their main mode of commuting to work, significantly lower than the automobile dependency in the city where 78% of residents rely on private vehicles for commuting [39], indicating local support for active travel modes. This neighborhood was selected for the School Street due to a high density of children living within a walking distance of the school, the traffic congestion at the school during drop-offs and pick-ups, and the surrounding grid-style street network that enabled the diversion of motorists. The plan for the School Street involved closing a 500-m section of road space for 30 min in the morning and 30 min in the afternoon, coordinated around school arrival and dismissal times. The School Street ran every school day during the 2021–2022 school year. Volunteers were recruited from the community to close the street and manage the School Street during closure periods. 

### 2.2. Methodological Approach and Initial Program Theory

This comparative case study used realist evaluation to explore how an outcome (the launch of a School Street) can be explained by the (in)actions of specific mechanisms in specific contexts. Realist inquiry is concerned with identifying the underlying mechanisms through which outcomes occur, and the contexts in which those mechanisms are triggered. Pawson and Tilley [40] describe these linkages as “context-mechanism-outcome configurations” (C-M-O configurations). A realist evaluation was therefore relevant to achieve the study’s objective to better understand what occurred during the implementation phase of a successful launch of a School Street and what may have been missing that would have led to an unsuccessful launch. 

The initial theory about the essential mechanisms for launching a School Street drew from MUEC’s (2015) participatory urban planning approach [33], since this was the framework that the implementers at both sites were using to guide their community engagement and mobilization efforts (Table 1). MUEC’s guide identifies six phases of participatory urban planning which then served as potential mechanisms in the initial theories the research team developed. Implementation of a School Street is considered the outcome of interest in this study. By studying the implementation in two cities, the unique value of this study is identifying the contextual elements needed to activate the mechanisms necessary for launching a School Street. 

An example of C-M-O configurations derived from our initial program theory (Table 1) is the following: a high level of acceptability of the intervention in the neighborhood (potential contextual element) could facilitate the establishment of a partnership with local actors (mechanism), which, in turn, could facilitate the launch of the intervention (outcome). Additionally, the baseline pattern of high numbers of motorized vehicles on the street (potential contextual element) could facilitate a common understanding of the “problem” amongst the local population (mechanism) and therefore facilitate the launch of the intervention (outcome). 

### 2.3. Data Sources

This realist evaluation employed qualitative data collection and analysis methods [41]. Specifically, the study drew material from various documents (Table 2) including meeting notes, emails, fact sheets, and Frequently Asked Questions documents produced by both the researchers and the implementers during the mobilisation phase. Research team members at each site developed dense chronological logs of key events in the years and months leading up to the planned School Street launch date of September 2021. Additionally, semi-structured key informant interviews were conducted to capture more detailed information from knowledgeable stakeholders about how and why the School Street interventions were launched or not. Key informants were defined as people who had a significant contribution or involvement in the School Street planning process such as municipal staff, implementation partners, school staff, parent leaders and volunteers. Purposeful sampling was used to recruit key informants, which involved selecting individuals that were especially knowledgeable about the phenomenon of interest, i.e., the implementation of the School Street [42]. Key informants were invited to participate through e-mail. Interviews were held via Zoom or by phone, based on interviewees’ preferences, and were conducted by researchers at each site. Interviews were conducted in both French and English based on the preference of the key interviewee. Five interviews were conducted with key stakeholders in Montreal, with an additional one stakeholder who opted to provide written responses through email. Five interviews were conducted with key stakeholders in Kingston. Ethics approval for this study was obtained from both the Queen’s University General Research Ethics Board and the Comité d’éthique de la recherche en sciences et en santé de l’Université de Montréal.

### 2.4. Analysis 

The analysis began by constructing a timeline of events in both cities using the documents and materials collected during the planning and implementation stages. This timeline was then used to identify the barriers faced and key facilitating factors while planning the School Street pilots. From these timelines or “stories”, the first round of C-M-O threads were developed using deductive content analysis with the MUEC framework as the initial theory (Table 1). This was followed by an inductive analysis of the stories to develop emergent contexts and mechanisms not captured in the MUEC framework. Context, Mechanism, M-O and C-M-O threads were then identified and compared between the two cities to identify similarities and differences. Member checking was done to confirm the identification of the context, mechanisms, and C–M–O linkages. Finally, a qualitative thematic analysis was conducted to code the data collected from the key informant interviews. Interview data were transcribed in full. Analyzed interview data were used to confirm contextual elements, mechanisms and C–M–O linkages identified from the stories as well as identify any new linkages. 

## 3. Results

The two-year implementation phase under study led to a successful launch of a School Street in Kingston and an unsuccessful one in Montreal. To better understand *how* and *why* outcomes differed between the two cities, we identified four mechanisms (Table 3) triggered or constrained by 11 contextual elements (Table 4). Using these contextual elements (C) and mechanisms (M) as well as C-M-O threads for each city, this section is organised as follows: first, we present and describe the four mechanisms key to a successful launch in Kingston, and only partially activated in Montreal. Then, we present and describe the contextual elements that activated or constrained the activation of the mechanisms. Finally, we use the contextual elements, the mechanisms and the outcomes (launch of the SS or failure to launch) and organize them as C-M-O threads to “tell the story” of what happened, how and why, in each city.

### 3.1. Mechanisms 

The four mechanisms identified in the analysis are: (1) “Partnership”; (2) “Legitimacy”; (3) “Collaborative governance”; and (4) “Community Mobilisation”. The dimensions of these mechanisms are presented in Table 3. Although the participatory urban planning approach (Table 1) was the researchers’ starting point for identifying potential mechanisms for School Street implementation, the mechanisms identified in the analysis were slightly different. For instance, in the cases under study, the implementers already had a solution (the School Street) to propose to communities. Creating a shared vision about the local issues and how the School Street can address them was therefore central to the “Partnership” and “Community Mobilization” mechanisms while “Exploring solutions” was not. 

### 3.2. Contextual Elements

Table 4 lists the contextual elements that facilitated or constrained the activation of the mechanisms depending on whether they were present or absent in each city. 

### 3.3. Activation of the Mechanisms 

The following paragraphs explain how the activation of the four mechanisms (M, Table 3) was facilitated or constrained by the contextual elements (C, Table 4) in each city. 

#### 3.3.1. Kingston 

In Kingston, the outcome of interest was achieved, and all four mechanisms were activated. 

The partnership mechanism (M1) was fully activated in Kingston as partnerships were formed between the implementers and the municipality, researchers and the school. The partnership with the municipality was the first relationship developed and was easily formed based on a previous working relationship between the implementers and municipality (C3), and the support from both the transportation manager (C6) and the elected city councillor (C4). 


*“We already have that relationship with the transportation department so that’s a positive because we’ve worked with them before on other projects and I think they wanted to help us and wanted to be supportive.”*
—Implementer 2

The partnership with the school was more challenging as the implementers were less familiar with the school principal, who had only been working at the school for a few months. However, the principal had a clear understanding of the problem and an awareness that innovation was needed (C9), thus, allowing for a trusted partnership to eventually form. The partnership with the principal was also facilitated by the unprecedented COVID-19 pandemic (C8) as the principal saw the pandemic as a motivation to participate in the School Street project as the closed street would allow greater space for physical distancing at busy arrival and dismissal times. Additionally, the implementers and researchers had relationships with the school community (C3) which may have assisted in the formation of a partnership as they were familiar with the school environment.


*“My initial reaction is always to be concerned with the safety of my students (…) So immediately. I was thinking well if it’s something that’s going to make my school more safe, I’m intrigued, and I want more information”*
—School Principal

Legitimacy (M2) was also achieved in Kingston facilitated by the previous relationship that implementers had with the community (C3); the implementers were residents of the area and therefore trusted by community members. Additionally, support from the local elected official (C4), the school principal (C5) and the municipality (C6) increased legitimacy as the community knew other trusted officials were involved. That many local residents also recognized the problem of congestion around the school (C9) lent further legitimacy to the proposed School Street.


*“Yeah, and then also I guess I’m involved as a resident. Since the school is right around the corner from my house. I know several of the people that live on the street, some of them I just kind of recognize.”*
—Implementer 2


*“[Safety] is very high priority for me, it was in my top 3 [issues] in my election campaign. So it’s top of mind for all of the residents of my district. There’s a constant problem with safety, especially for children (…) So my initial reaction to the project was that it’s great, get it going as soon as possible”*
—Elected official


*“Well, certainly, the principal was cooperative and you know quite actively promoting it as opposed to just kind of you know, holding their nose and letting it happen and that helped a lot early on.”*
—Implementer 1


*“Going around knocking door to door on the streets, most of the people we encountered at the time (…) were happy about it or, like, ‘oh, this is great’ or ‘we really need this’ or ‘this is important’. We got quite a few of those responses.”*
—Implementer 2

In Kingston, collaborative governance (M3) was achieved between the municipal staff, the school, implementers and the researchers. Collaborative governance was facilitated by the previous working relationship (C3) between the implementers and the municipality which made collaboration easier. The support from the school principal (C5) and the municipality (C6) also enabled collaborative governance as they were both willing to dedicate staff time to advancing the approval of the intervention. 


*“Luckily we had enough goodwill with the council that they knew, and I think transportation services [department] knew, that we had done one thing and done it quite well. And so we had a bit of a track record with the municipality”*
—Implementer 1

Once collaborative governance was formed, the partners worked collaboratively to mobilize support for the project in the community (M4). In Kingston, many students attending the target school were already using active travel (C7) and therefore many parents were aware of the importance of road safety and active travel to school (C9). This facilitated community mobilization with parents as implementers did not need to provide education on the importance of active travel and parents easily understood the project.


*“And so I guess because I felt more connected to [the project]. I think because there’s a connection in our school community to the actual project.”*
—School Principal

These contextual elements in Kingston, in conjunction with the school principal’s support (C5) and their willingness to send communications to parents, greatly facilitated the activation of community mobilization (M4). 

#### 3.3.2. Montreal 

Despite the outcome in Montreal (failure to launch the School Street), some of the mechanisms described in Table 3 were partially activated during the implementation phase, which led to significant developments toward a successful launch between 2019 and 2021. For instance, a “Partnership” (M1) was successfully established with the municipality, which ultimately led to a change of bylaw at the municipal level to allow the School Street to run. However, the absence of a committed partnership with the school and the lack of a collaborative work dynamic between partners negatively affected the legitimacy of the project in the long run. The inability to activate legitimacy (M2) and collaborative governance (M3), and to only partially activate partnership (M1) and community mobilization (M4), explains the failure to launch the School Street in Montreal. 

First, the built environment (C1) surrounding the projected School Street as well as a strong awareness of issues regarding road safety (C9) at the municipal level facilitated a quick establishment of a partnership with the municipality (M1). Indeed, the School Street was planned to be a narrow one-way street bordered by two schools, with a high volume of motorized traffic (C7) during the school’s drop-off and pick-up. This situation had already been identified by the municipality as problematic regarding children’s safety:


*“We stand for security around schools (...), and well, it’s a disaster around these schools (...) It’s so unsafe”*
—Elected Official

Although the partnership with the municipality was quickly established, a partnership with the school community was never fully activated (M1). Ongoing dialogue with the school community took place over the course of a one-year period with the objective for all partners to move towards a common understanding of the importance of the project and its projected benefits. One of the two schools expressed its willingness to be involved as a partner, however, the school board and the principal of the second school expressed persistent concerns about the school’s roles and legal responsibilities, as well as doubts about the project’s feasibility, ultimately preventing them from supporting it (C5).

The legitimacy (M2) of the project was partially activated, as the municipality trusted the implementers, likely due to the already established partnership between the university-based research team and the non-profit organization (the implementers) (C3), who were co-leading the project:


*“A good move is to have a non-profit organization (…) with a university, especially for liability insurance, support, etc… it’s the best of both worlds, there really is an added value”*
—Elected Official

Legitimacy (M2) was primarily constrained by the lack of buy-in from the school principal (C5) at the second school, who did not demonstrate any willingness to be involved in the project:


*“We have a lack of enthusiasm from management, from the principal to say it clearly, and also from a few other people on the board, that it will annoy people who come by car (…). So there was a very OK official speech, everyone is in favour of virtue, active mobility, but in practice, I would say that the principal was not in favour of the project.”*
—Parent

The school board also believed that the feasibility of the project was undermined by the COVID-19 pandemic (C8). This perceived lack of feasibility constrained legitimacy (M2):


*“During the pandemic, management worked every weekend and until 10 pm in the evening to deal with COVID cases. So to start a new project in this context was a little utopian.”*
—School Board Representative

Additionally, the municipality did not openly voice its involvement in the School Street when interacting with the public or other stakeholders, which might have contributed to undermining the legitimacy of the project to the community.


*“At no time did we have a clear message, a public and clear message from the borough saying: We want this project…. In my opinion, it has contributed to undermining our legitimacy.”*
—Implementer

The municipality was supportive of the intervention, and the project had a clear buy-in from an influential elected official (C4), which facilitated the partnership (M1). The lack of a “champion” manager inside the municipality administration (C6), however, seemed to constrain a collaborative governance (M3) dynamic from taking place between the implementers and the municipality:


*“I think the borough was very, very in favour of the intervention as an institution, but I don’t think we had the right actor to promote this internally and then externally ”*
—Implementer 

Collaborative Governance (M3) was also constrained as the second school continued to be neutral (C5) towards the intervention, and stakeholders’ responsibilities for elements of the project seemed misunderstood. Implementers were expecting the municipality to play a leadership role with the schools and the community, but the municipality felt that this was the role of the implementers:


*“The partnerships with the schools, that’s something that I can’t do because I’m not the implementer of the project. I can just be there in support but it’s not my project (…). If you don’t have the schools, you don’t have a project, so you should have secured it first.”*
—Municipality

Community mobilization (M4) was partially activated in Montreal with a group of parents. Parents seemed to be aware there was a problem with traffic congestion around the school (C9) and were therefore receptive to the idea of a School Street. 


*“There was (…) a good group of parents there who are interested in these things and who were very supportive and who said well what can be done to push this so that the management of the school accepts the project? ”*
—Parent

However, this mechanism was constrained by the low levels of active school travel in the community (C7) and the lack of assistance from the school principal in communicating with parents about the benefits of active school travel (C5). Without the buy-in of the school principal, in particular, the project lacked legitimacy. This school’s lack of willingness to participate created delays in the action plan with the other school, which eventually disengaged:


*“This ambiguity and (…) their questions contributed to a loss of trust from the other partner. That’s what we experienced, we gradually saw it happen, the other partners gradually disengaging.”*
—Implementer

Taken together, the partial activation of M1 and M4, and the failure to activate M2 and M3, led to the decision by the implementer in Montreal to halt the pursuit of a School Street at this location for the 2021/2022 school year which led to a non-launch.

#### 3.3.3. Context–Mechanism (C–M) Linkages 

Based on the analysis of the mechanisms in each school site, four C–M linkages were identified (Table 5) that explain why the School Street successfully launched in Kingston and not in Montreal. 

The lack of support from the school principal (C5) in Montreal constrained the activation of each of the four C–M linkages. In Kingston, the school principal quickly signed on as a partner and allowed for the activation of legitimacy, collaborative governance and community mobilization as they publicly displayed their support and assisted with tasks at each stage of the project. This enthusiastic and supportive response never materialized in Montreal, which created challenges in fully activating all four mechanisms. Another key contextual difference between the city sites was the level of support from a manager or leader within the municipality (C6); without such support, the legitimacy of the project and the opportunity for collaborative governance were both severely constrained in Montreal. 

## 4. Discussion

### 4.1. Key Findings 

Children are increasingly spending more of their free time indoors as parents give fewer licenses for children to play and move about independently in their communities [43]. The lack of experience playing outdoors and travelling independently restricts children’s ability to develop resiliency and skills to help cope with challenges later in life [43,44]. School Streets are interventions that aim to provide child-friendly public spaces prioritizing children’s mobility over private automobiles. Little is known, however, about how and why these interventions are launched in some places but not others, suggesting a lack of programme theory. 

The objective of this study was to investigate the mechanisms and contextual elements that led to the successful implementation of a School Street intervention in one community and to better understand why the intervention failed to launch in another. The findings revealed that four mechanisms—partnerships, legitimacy, community mobilization, and collaborative governance—were fully activated by a confluence of several contextual elements in Kingston, where the School Street was successfully launched. In Montreal, where the School Street failed to launch, partnerships and community mobilization were partially activated, while legitimacy and collaborative governance failed to be activated at all. The most notable contextual element that explained the failure to launch was the lack of a supportive school principal to champion the intervention as well as support from a manager or leader within the municipality. 

The analysis highlighted that synergism between the four mechanisms is needed to successfully launch a School Street. Legitimacy is at the heart of that synergy: without this mechanism, the processes of involving partners, collaborating and mobilizing the community seemed constrained, and without a partnership, collaborative governance or a mobilized community, legitimacy was negatively affected. Indeed, other researchers have argued that legitimacy is at the core of social innovation [45,46,47]. Social innovations are new social practices created from collective, intentional, and goal-oriented actions aimed at prompting social change and potentially generating socio-political transformations [45,46]. School Streets in the Canadian context are social innovations challenging the status quo and calling for a paradigm shift to create more child-friendly cities. 

### 4.2. Shoring Up Support

The importance of the school principal’s support for the successful implementation of a program is well-documented in this paper as well as in the active school travel literature, where programs with similar objectives to a School Street, including Walking School Buses, Safe Routes to School and Remote Drop-offs, found that a lack of school support was a major barrier for successful implementation of the programs [34,48,49,50,51]. Research has shown that if active school travel is not recognized as a major priority for the school, then implementation becomes over-reliant on external inputs [34] and that externally driven interventions face more barriers to implementation [52]. While both teams possessed some degree of “insider status” with the school communities, this status was less important than the school principal’s awareness of the benefits of School Streets, their interest in increasing active school travel, and most importantly, their willingness and capacity to take on key tasks to support the implementation of the intervention. If the intervention must be externally driven, the findings from this study and elsewhere [34], suggest that assessing school readiness at the outset, and finding ways to align the intervention goals with the school administrators’ priorities (e.g., safety) are critical to shoring up support from this critical stakeholder. 

This study also demonstrated that simply forming a partnership with an entity (e.g., a municipal government) is not enough to trigger a social innovation like a School Street. In the case of the School Street in Montreal, the choice of the street between two primary schools was the express desire of the municipality. The high levels of traffic on the chosen street, along with the long-standing desire amongst elected officials in the borough to convert the street into a pedestrian way, drove the choice for our School Street location, despite our team’s fears that working with two separate schools might create problems. Research on Safe Routes to School programs has shown that the most productive partnerships include a diverse range of actors (e.g., students, parents, schools, municipal councillors and policy developers) working collaboratively. Such arrangements not only increase collective capacity to problem-solve, but also lend greater credibility to the intervention [49]. Cross-disciplinary partnerships may also help legitimize School Street initiatives, which this study suggests is necessary for successfully launching a School Street.

### 4.3. Implications of Street Rebalancing for Creating Child-Friendly Cities

For roughly a century, the planning and design of North American cities have privileged the automobile at the expense of the needs and desires of children. As such, individuals and groups working to create child-friendly cities through street-based innovations need to be prepared to face resistance from the people, structures, and systems that support and uphold the status quo. In the context of School Streets, the findings from this study demonstrate how crucial it is to create productive partnerships with elected officials, school staff, municipal staff and parents to ensure that the benefits of re-designing streets for children are broadly understood and supported before initiating the School Street mobilization process. Without buy-in from a diversity of stakeholders, implementers will face insurmountable challenges in successfully launching and maintaining a School Street. With automobility so deeply entrenched in North Americans’ psyches, disgruntled individuals and groups will inevitably express their discontent with socially innovative projects like School Streets. Thus, project partners need to be strong champions at the outset of the project and remain steadfast in the face of potentially intense opposition. 

As School Streets inconvenience the motorist in favor of children, there will be those who contest the purpose of the School Street as our everyday lives have been structured around the car [53]. Therefore, to prove to naysayers that the benefits outweigh the costs, it may be worthwhile to pilot innovative street-rebalancing projects in communities with already high baseline levels of children’s use of the street (e.g., for play and/or active school travel)—the “low-hanging fruit”. In Kingston, the participating school community was not the most in need of a School Street relative to other sites in the city; however, its success as a full-year pilot is likely partially explained by the high levels of active school travel in the community that helped activate the identified mechanisms. Indeed, the success at this one site has triggered interest in establishing additional School Streets across the city and in other municipalities, suggesting that seeing is believing when it comes to street rebalancing initiatives. 

Child-friendly cities are not only focused on providing safe and accessible outdoor spaces but also need to provide opportunities for children to participate in their community and feel included [54,55]. School Streets offer opportunities not only for children to more safely use and access public space but can provide opportunities for children to be greatly involved and engaged in a community development project. Children are traditionally left out of the community engagement process as youth are not fully accepted as active “citizens” and therefore planners and public health practitioners assume they are not able to fully process information to make practical decisions [56]. Yet it has been found that involving children in planning communities provides health and well-being benefits to children and promotes understanding of and enthusiasm for community development projects [57,58,59].

Despite the known benefits of involving children in planning at the partnership level, this project, along with many other child-centered projects, did not include children as project partners. In initial plans, children were identified as key stakeholders, however, the COVID-19 pandemic and the associated school closures made accessing and engaging children much more challenging. Nevertheless, the fact that children were excluded from a project aimed at improving their experience in a city could likely be a contributing factor as to why the Montreal School Street failed to launch. In projects where children’s input is sought and included in project planning, there is a greater compulsion for the project to be fully implemented, likely due to children’s enthusiasm developed through the engagement process [56,60]. Perhaps the school principal in Montreal may have felt more pressure to support the School Street had their students been excited about and anticipating the start of the School Street, leading to the successful launch of the intervention. The case in Kingston displays that School Streets can be launched without a partnership between children and implementers, however, if implementers are focused on advocating for children’s rights to the city, then children ought to be included.

### 4.4. Study Limitations and Future Research 

This study was limited in its examination of only two School Street pilots. The analysis of additional successful and failed pilots would enable us to better determine whether our list of mechanisms and contextual factors is comprehensive and whether some C–M links are more important than others. Additionally, this study focused on identifying the mechanisms and C–M linkages that led to the successful launch of a School Street in Kingston and then determining which mechanisms were missing in Montreal. Therefore, this study did not explicitly study the mechanisms leading to failure as this was beyond the scope. Finally, future research could interview a greater number and range of stakeholders to ensure the perspectives of all those involved are captured.

## 5. Conclusions

Prioritizing children’s needs for outdoor play and independent mobility can be difficult in the automobile-dominated landscapes of North America. As many Canadian parents rely on motorized travel to get their child to and from school, children are being deprived of critical opportunities for social, physical, and psychological development. School Streets can offer low-cost solutions for children to reclaim the street for play and mobility and for cities to become more child-friendly, but numerous barriers are preventing these innovations from being implemented. Indeed, the paradigm shift required in municipal policy making, schools’ vision, and communities’ priorities to operationalize School Streets is difficult to initiate. This study found that to successfully implement School Streets, school boards, principals and municipalities must be willing to prioritize child well-being and development over the interests of motorists. This will require brave leadership from multiple sectors who are willing to challenge the status quo. Furthermore, these social innovators need to form trusting relationships with their target communities, including children, to ensure that the School Street is designed around their needs and that the community can understand the benefits from the start. The findings from this study offer important insights into the mechanisms and contextual elements needed to successfully pilot innovative active school travel projects. That is until child-friendly urban environments become more widely accepted amongst the public. Future research on combatting automobile culture in North America is needed to more successfully advocate for children’s right to the city. 

## Figures and Tables

**Table 1 ijerph-19-11555-t001:** Initial framework for mechanisms about how and why the School Streets might launch as adapted from the Participatory Urban Planning guide from MUEC.

Phases of Participatory Urban Planning (Potential Mechanisms)	Actions to Activate the Mechanisms
Get started: Establish a partnership with local actors and define an action plan	Co-define a partnership agreement Define the scope of the intervention Define the objectives Define everyone’s roles and responsibilities Establish a timeline Determine human, material and financial resources
Understand: Undertake a diagnostic portrait of the space to promote a common understanding of the issue(s)	Creation of and consultation with a steering committee Asset mapping Exploratory walks Consultation booth Surveys
Explore solutions: Identify design possibilities that meet the needs of the community	Workshop Bring together all stakeholders to imagine solutions adapted to the identified needs Promote the establishment of a dynamic collaborative work between actors
Decide: Validate with the different actors the developed solutions and enrich them	Scenario validation workshop Working session with municipal professionals
Act: Start implementing the solutions and make commitments	Pilot the project or temporary measures Creation of a monitoring committee

**Table 2 ijerph-19-11555-t002:** Summary of Data Sources.

Source Type	Source		Montreal	Kingston
Documents (February 2019–November 2021)	Meeting minutes	With municipality	5	2
		With school	7	2
		With multiple stakeholders	1	4
		With research team and implementers	32	5
	Supporting documents	Mobilization protocol, progress report, letters to residents, FAQ sheets, municipal report	2	6
	Researchers’ notes	Chronological log of important events	2	1
Interviews (September 2021–May 2022)	Municipality representatives (staff or elected)		2	1
	School representatives (staff or parent council)		3	1
	Implementers (paid or volunteer)		1	2

**Table 3 ijerph-19-11555-t003:** Mechanisms identified for a successful School Street launch.

**M1. Partnership**		
** *Description* **	** *Actions involved* **	** *What it looked like when activated* **
Local actors coming together with a shared vision that a School Street is a tool for increasing safety, free play and active transportation around the school.	Holding meetings to present, explain and discuss the project, defining the scope of intervention; and establishing an action plan	Municipality, implementers, the target school and the researchers formed a working relationship and were all in support of a common goal (launch of the School Street).
**M2. Legitimacy**		
** *Description* **	** *Actions involved* **	** *What it looked like when activated* **
The process of building trust within the community and between the partners, creating trust that the implementer(s) can implement and manage the School Street.	Displaying the implementers’ expertise to lead the intervention, emphasizing the team’s understanding of the community, representation and support from key institutions (municipal staff, elected officials and the school)	Community understands the intervention and trusts the implementers, the municipality, school and researcher’s involvement is highlighted in information guides.
**M3. Collaborative Governance**		
** *Description* **	** *Actions involved* **	** *What it looked like when activated* **
Collaborative governance forms from initial partnerships and is reached when conflicts are resolved collaboratively with partners, partners are in constant communication and there is a delegation of tasks between partners	Delegating clear and mutually agreed upon tasks and roles amongst the partners, establishing effective communication channels, and resolving conflicts and issues through negotiation with partners.	All partners have their own tasks and complete them within the agreed timeline, the communication between partners is frequent and fluid.
**M4. Community Mobilization**		
** *Description* **	** *Actions involved* **	** *What it looked like when activated* **
The process of informing the community and providing opportunities for discussion and feedback and inspiring people to be supportive and involved.	Creation of a shared vision in the community and a mutual understanding that a School Street is a practical solution to local issues.	School parents and residents were given the opportunity to learn about the intervention, provide feedback and get involved in the intervention.

**Table 4 ijerph-19-11555-t004:** Contextual elements identified in analysis.

Context	Description
C1. Built environment	The residential density in the school community, the street layout around the school site and street design features.
C2. Regulatory process	The approval processes in place from both the municipality and the School Board to implement the School Street.
C3. Previous history between actors	The existing working or personal relationships between the local actors.
C4. Supportive Influential elected official	The presence of an influential elected official supportive of the School Street.
C5. Involvement of a “champion” school principal	A school principal who was supportive of the School Street.
C6. Involvement of a “champion” within the municipality	Municipal staff member was supportive and actively working to advance the intervention.
C7. Baseline patterns of motorized and non-motorized street use	The levels of active travel in the school community compared to levels of car travel prior to the School Street.
C8. Stable environment	The working and school environment were conducive to the start of a new project
C9. Level of awareness of the “problems” in the community	The level of understanding of the “problem” identified in the community that School Streets will address (i.e., road safety, lack of active travel, decreased independent mobility)

**Table 5 ijerph-19-11555-t005:** Context–mechanism linkages for the implementation of School Streets in Kingston and Montreal.

Context-Mechanism Linkage	Activated in Kingston?	Activated in Montreal?
C-M 1. Partnership between municipal staff, implementers, researchers and school stakeholders was activated through previous work history between actors (C3), presence of a champion school principal (C5) and a champion in the municipality (C6) and/or an elected official (C4).	Yes	Partial, Missing C5
C-M 2. Legitimacy was activated by a previous relationship with the community (C3), public support from an influential elected official (C4) and/or a manager at the municipality (C6), public support from the school principal (C5) and a stable working environment between partners (C8).	Yes	No, missing C3, C4, C5, and C8
C-M 3. Collaborative governance was activated through the support from the school (C5) and the municipality (C6), and previous working relationships between partners (C3).	Yes	No, missing C5, and C6
C-M 4. Community Mobilization is activated through support from the school principal (C5), high levels of active transportation pre-intervention (C7) and/or an awareness of the problem of road safety around the school (C9) and high availability of human resources nearby the school to support the intervention (C9).	Yes	Partial, missing C5

## Data Availability

The data are not publicly available to protect the privacy of interviewees.

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
