# Peer review of "A Tale of Two Cities: Unpacking the Success and Failure of School Street Interventions in Two Canadian Cities"

_ijerph, 2022, doi:10.3390/ijerph191811555_

Round 1

Reviewer 1 Report

General Comments

This paper evaluates the contextual elements and mechanisms that support or hinder successful implementation of a change to the built environment called “School Streets” that could promote children’s health. It focuses on a high impact, tactical urbanism strategy that could increase public awareness of the safety hazards to children that have been designed into communities whose transportation networks prioritize single-occupancy vehicles over all other modes of transportation. The article is well researched and well written, with a logical flow of ideas. Its analysis focuses on the gap in knowledge around why some school streets programs launch successfully, and others do not. Section 3. Results and 4. Discussion - Shoring Up Support are particularly strong.

Comments by Section

1. Introduction – Evaluating School Street Implementation using Critical Realism (lines 93-113): Please expand this section to explain more fully the researchers’ rationale for using a critical realism approach compared with another approach; their epistemological, ontological, and reflexivity assumptions entering the project; and, how critical realism theory influenced (or didn’t influence) to their decision to use participatory urban planning to select the intervention sites and perform the intervention itself.

2.1. Materials and Methods – School Street Intervention Settings: 

(lines 119-123): Please add a short literature review (1-2 sentences) placing participatory urban planning within the context of other studies that have considered why a planning intervention was or was not successful.

 (lines 131-132 and lines 143-145): The comparison of car ownership/usage is not consistent across the two neighborhoods. It would be helpful to understand what percentage of the adults in the Montreal neighborhood report using a private vehicle as their main mode commuting to work (compared with Montreal as a whole), so that readers can compare driving culture between the two sites. Montreal has a reputation of higher multi-modal transportation use than many other parts of Canada, but that may not be the case in the neighborhood where the intervention was sited.

Table 1 (line 167): Why is the street grid not considered as part of the context, since it is an essential component to the successful diversion of traffic? If it is implied in “baseline patterns of motorized and non-motorized street use,” I would suggest editing this line to clarify what is meant by this aspect of the context.

I have a similar comment about the 3rd item under context. More than “acceptability of the intervention,” it seems like researchers might want to understand the extent to which local residents, key informants, children, and parents consider the traffic congestion around schools as a problem. 

2.3 Data Sources:

(lines 182-195): Please describe the participant recruitment approach and inclusion/exclusion criteria. Random sampling? Convenience sampling? Snowball? How did researchers guard against selection bias that would favor interviews with participants who were supportive of the school street intervention and less likely to share potential barriers to its successful implementation?

4. Discussion. Key Findings: (lines 428-430) I was not convinced that the lack of support from a principal and municipal manager were the primary contextual elements explaining the failure for the school street intervention to launch in Montreal. I wondered about the low levels of active transportation to school before the intervention. It is possible that caregivers were not traveling from their home to pick up their children at the end of the day. If they commuted by car, then being asked to drive home, park, and then walk to pick up a child at school would potentially compromise their work schedule. I also wondered about the decision to try out the Montreal pilot on a street with two schools. One of the principals was supportive of the program; one was not. Perhaps the decision to implement the program on a street that required buy-in from two sets of principals, school boards, and administration was overly ambitious on the researchers’ part.

Author Response

Please see the response to Reviewer #1's revisions in the attached Word Document. 

Reviewer 2 Report

The work is very interesting and actual, in the context of sustainable cities and the environment (SGD11.Make cities and human settlements inclusive, safe, resilient and sustainable). It is very interesting to carry out studies like this so that the policies of the cities change to a safer and more sustainable environment.

The work is very well documented, with a high number of articles between the references.

There are some questions, which may not be decisive for the study, but which have arisen when reading it:

- How were the volunteers who participated in the management of the streets selected in each of the periods, in both cities? What are their characteristics? Are they parents, teachers, neighbors?

- In table 1, within the actions to activate the mechanisms, there is talk of financial resources. How much has it cost? What type of financing has been necessary?

- Is there quantified data on the success of the project? For example, children positively affected, number of families satisfied with this policy.

- Lines 349-350: “The school board also believed that the feasibility of the project was undermined by 349 the COVID_19 pandemic (C8). This perceived lack of feasibility constrained legitimacy 350 (M2)”, in the Montreal case. In Kingston, did COVID_19 not affect the Project?

As a possible help in the success of these actions:

- Have you tried to include companies related to children, physical activity or the environment to sponsor this activity? Perhaps, with companies as partners, some schools would be more willing to collaborate.

Author Response

Please see our responses to Reviewer #2's comments in the Word document attached. 
